# OpenReview forum: "MJ-Video: Benchmarking and Rewarding Video Generation with Fine-Grained Video Preference"
_NeurIPS.cc/2025/Conference — NeurIPS 2025 spotlight_

### Official Review · Reviewer_XJP5 · 2025-06-02

**Clarity:** 3
**Significance:** 3
**Originality:** 3
**Rating:** 5
**Confidence:** 4

**Summary:**

This paper introduces MJ-Bench-Video, a benchmark for evaluating text-to-video generation across five key aspects: alignment, safety, fineness, coherence and consistency, and bias and fairness. The benchmark includes 5,421 data entries annotated with 28 preference criteria. To leverage this dataset, the authors propose MJ-Video, a Mixture-of-Experts (MoE) reward model that accounts for the different aspects and preference criteria. The authors demonstrate that MJ-Video outperforms existing vision-language models and reward models, in both fine-grained and overall video preference judgment. Additionally, they demonstrate that they can leverage the reward model to improve existing video generation models.

**Questions:**

1. It is unclear whether the reward model can generalize to better models. Can it provide reliable preference judgments for more performant models such as SORA, or the latest version of ltx-video?

**Ethical Concerns:**

["NO or VERY MINOR ethics concerns only"]

**Final Justification:**

I will keep the score as is, the strength and weaknesses justify the score.

**Limitations:**

yes

**Paper Formatting Concerns:**

ok

**Quality:**

3

**Strengths And Weaknesses:**

Strengths:
1. The proposed benchmark and reward model are a useful contribution.
2. The improvements over strong baselines such as GPT-4o are solid.
3. The experimental setup is extensive.
4. The usage of the reward model and the gains when compared to other reward models are solid.
5. The ablation study that supports the architecture design choices for the reward models are solid.

Weaknesses:
1. The scale of the data, which includes only 5K training examples is not so vast. It may be, that with a much larger dataset, the architecture design choices will be less useful. Moreover, the usage of 28 different criteria can make it harder to scale such a preference dataset in the future.
2. It is unclear whether the reward model can generalize to better models. E.g. can it provide reliable preference judgments for more performant models such as SORA?

---

> ### Author Rebuttal · Authors · 2025-07-31
>
> Thank you for your constructive advice! We would like to mitigate your concerns by the following point-to-point response.
>
>
> > The scale of the data, which includes only 5K training examples is not so vast. It may be, that with a much larger dataset, the architecture design choices will be less useful. Moreover, the usage of 28 different criteria can make it harder to scale such a preference dataset in the future.
>
>
> **A1:** First, to validate the effectiveness of the MoE architecture, we replace it with a dense architecture while using the same fine-grained training data. We introduce two variants: (1) MJ-VIDEO-Aspect, which only uses single MoE layer for 5 aspects. (2) MJ-VIDEO-Dense, which  applies a standard ranking loss across each criteria, followed by averaging. These two variants use the same input information of MJ-BENCH-VIDEO as MJ-Video for training. The following results demonstrate that our hierarchical MoE architecture can more effectively utilize fine-grained information than dense models.
>
>
> | **Model**                | **MJ-BENCH-VIDEO** | **Safesora-test** | **GenAI-Bench** |
> | ------------------------ | ------------------ | ----------------- | --------------- |
> | MJ-VIDEO-DENSE       | 63.09              | 57.87             | 68.77           |
> | MJ-VIDEO-ASPECT      | 66.17              | 67.29             | 63.15           |
> | **MJ-VIDEO** | **68.75**          | **64.16**         | **70.28**       |
>
>
>
> Second, although we cannot scale our data within the limited rebuttal period, we conduct scaling experiments by gradually increasing the training set size — from 100, 1000 to 5000 (full). As shown below, we find that dense architecture has some advantages with limited training data, while MoE architecture starts to beat dense model after using 1000 data. This dicates that the adopted MoE architecture is a better candidate for scaling.
>
>
> | **Model**                      | **MJ-BENCH-VIDEO** | **Safesora-test** | **GenAI-Bench** |
> | ------------------------------ | ------------------ | ----------------- | --------------- |
> | DENSE-100         |  **54.21**       | **51.40**       | 50.08     |
> | MoE-100         | 49.59        | 50.35       | **51.04**     |
> | DENSE-1000        | 59.41       | **56.19**       | 62.38     |
> | MoE-1000 | **61.27** | 52.18 | **64.65**|
> | DENSE-5000 | 63.09              | 57.87             | 68.77           |
> | MoE-5000       | **68.75**          | **64.16**         | **70.28**       |
>
> To ensure high quality and capture nuanced human preferences, we design MJ-BENCH-VIDEO with 28 fine-grained criteria. However, it can be easily extended or simplified by reducing the fine-grained annotations to aspect-level annotations (i.e., reducing from 28 to 5).
>
> ---
>
> > It is unclear whether the reward model can generalize to better models. Can it provide reliable preference judgments for more performant models such as SORA, or the latest version of ltx-video?
>
> **A2:** To further demonstrate the potential and generalization of our MJ-VIDEO, we test our model on the generated 100 videos by Sora and latest version of ltx-video[1] as follows:
>
> | **Reward Model**      | **Sora (%)** | **ltx-video (%)** |
> |-------------------|----------|---------------|
> | Video-Score       | 73       | 66            |
> | Human Annotators  | 85       | 80            |
> | **MJ-VIDEO**          | **79**       | **72**            |
>
>
> We can find that MJ-VIDEO perform consistently better than existing video rewarding model, showing promising generalization performance on high quality videos.
>
> [1] HaCohen, Yoav, et al. "Ltx-video: Realtime video latent diffusion." arXiv preprint arXiv:2501.00103 (2024).

---

### Official Review · Reviewer_WCuD · 2025-06-28

**Clarity:** 3
**Significance:** 3
**Originality:** 2
**Rating:** 4
**Confidence:** 3

**Summary:**

This paper proposes a multi-objective text-to-video evaluation benchmark, MJ-BENCH-VIDEO, for comparing different text-to-video reward models. The dataset is constructed by filtering or regenerating existing text-to-video data, followed by human preference annotations across 28 fine-grained criteria spanning five aspects: Alignment, Safety, Fineness, Coherence & Consistency, and Bias & Fairness. Based on this benchmark, the authors train a MoE text-to-video reward model, MJ-VIDEO, using the training split of MJ-BENCH-VIDEO. MJ-VIDEO achieves strong results on three benchmarks. In addition, this paper show that integrating MJ-VIDEO into an RLAIF loop to fine-tune VideoCrafter-v2 leads to improved human-rated video quality.

**Questions:**

For the baselines, how were the fine-grained scores aggregated into aspect-level or overall scores?

What is the inter-annotator agreement for each evaluation aspect?

**Ethical Concerns:**

["NO or VERY MINOR ethics concerns only"]

**Final Justification:**

The new ablation study results presented in the rebuttal are convincing. I have raised my score accordingly.

**Limitations:**

yes

**Quality:**

3

**Strengths And Weaknesses:**

Strengths

1. MJ-BENCH-VIDEO provides dense and diverse human annotations (5 aspects, 28 criteria), which will be valuable for fine-grained reward model research.

2. MJ-VIDEO achieves strong performance across multiple aspects and benchmarks.

3. When used as a reward model in an RLAIF loop, MJ-VIDEO improves human-rated video quality, demonstrating real downstream benefits.

Weaknesses

1. The overall approach follows a standard recipe for training and evaluating reward models. No novel algorithmic insight is introduced.

2.  MJ-VIDEO is trained on the same benchmark it is evaluated on, while the baselines are mostly zero-shot. The comparison would be more convincing if conducted under similar training conditions or tested on out-of-domain data.

3. For the MOE model, although the stated goal is to *train specialized experts per aspect*, the implemented scoring layer predicts all 28 criteria simultaneously and only applies scalar weights during aggregation. This weakens the claim of aspect-level specialization.

4. In the ablation study, the ablated models seem to use less annotation information than the full model, making the comparison unfair. For example, the “w/o Aspect MOE” variant directly predicts the overall score and does not appear to be trained on fine-grained annotations.

---

> ### Author Rebuttal · Authors · 2025-07-30
>
> Thank you for your thoguhtful feedback on our paper! We would like to clarify and mitigate your concerns by the following point-to-point response.
>
>
> > The overall approach follows a standard recipe for training and evaluating reward models. No novel algorithmic insight is introduced.
>
> **A1:** Our primary contributions are twofold: (1) the creation of fine-grained annotations specifically tailored for text-to-video alignment, and (2) the design of a hierarchical Mixture-of-Experts (MoE) architecture that performs multi-level routing—both at the criteria level and the aspect level—to serve as a reward model for video alignment. Here, this MoE framework provides a generalizable solution for building reward models in other domains with fine-grained preference labels, demonstrating how to leverage such annotations effectively. We believe this represents a meaningful algorithmic contribution to the community.
>
>
> ---
>
> > MJ-VIDEO is trained on the same benchmark it is evaluated on, while the baselines are mostly zero-shot. The comparison would be more convincing if conducted under similar training conditions or tested on out-of-domain data.
>
>
> **A2:** Table 2 already includes evaluations on two widely used out-of-domain datasets — SafeSora[1] and GenAI-Bench[2] — where MJ-VIDEO consistently outperforms baseline methods, highlighting its strong generalization capability. We further fine-tuned the open-source VideoScore reward model on the same training split of our MJ-BENCH-VIDEO dataset and compared it with our MoE-based model. MJ-VIDEO demonstrates superior performance under identical training conditions. Importantly, MJ‑BENCH‑VIDEO itself is a major contribution: it can be used to train future reward models with fine-grained scores for even stronger performance. Our paper shows the potential of such fine-grained annoations for text-to-video alignment.
>
> | **Model**                | **MJ-BENCH-VIDEO** | **Safesora-test** | **GenAI-Bench** |
> |--------------------------|--------------------|-------------------|------------------|
> | VideoScore           | 58.47          | 55.33         | 69.14        |
> | VideoScore-Finetuned | 64.18          | 60.07        | 70.12        |
> | **MJ-VIDEO**             | **68.75**          | **64.16**         | **70.28**        |
>
> ---
>
> > For the MOE model, although the stated goal is to train specialized experts per aspect, the implemented scoring layer predicts all 28 criteria simultaneously and only applies scalar weights during aggregation. This weakens the claim of aspect-level specialization.
>
> **A3:** Our MoE design is critical at inference time, as the model must automatically determine which aspects are most relevant for each video **without prior knowledge of the video’s focus** (e.g., safety). The router dynamically assigns weights to aspect-specific experts to produce an overall score.
>
> When the evaluation domain is known in advance, we can also utilize the aspect-level score (e.g., from the safety expert) in MJ-VIDEO to leverage domain-specific knowledge. Thus, we demonstrate this on our safety-focused subset as follows. Interestingly, the results show that the overall MJ-VIDEO score closely matches that of the dedicated safety expert, validating the effectiveness of our router. Further, it is easy to extract single expert for domain specific fine-tuning, since every expert does not rely on each other and can be fully independent.
>
>
> | **Model**             | **Accuracy** | **F1 Score** | **Strict Match** |
> | --------------------- | ---------------- | ---------------- | -------------------- |
> | Safety Expert  | 87.50            | 81.84            | 83.33                |
> | Full Model | 85.94            | 80.23            | 81.67                |
>
> ---
>
> > In the ablation study, the ablated models seem to use less annotation information than the full model, making the comparison unfair. For example, the “w/o Aspect MOE” variant directly predicts the overall score and does not appear to be trained on fine-grained annotations.
>
> **A4:** To rigorously test the necessity of the MoE structure under equal annotation inputs, we implement a dense judge baseline that ingests all fine‑grained preference labels and applies a standard ranking loss across every aspect before averaging. The comparison is as follows:
>
> | **Model**                  | **MJ-BENCH-VIDEO** | **Safesora-test** | **GenAI-Bench** |
> | -------------------------- | ------------------ | ----------------- | --------------- |
> | MJ-VIDEO (MoE)              | **68.75**              | **64.16**             | **70.28**           |
> | Dense Model | 63.09              | 57.87             | 68.77           |
>
>
> Under these controlled conditions, our hierarchical MoE model still outperforms the dense judge, demonstrating its superior ability to leverage fine‑grained annotations via adaptive routing and weighting.
>
> ---
>
> > For the baselines, how were the fine-grained scores aggregated into aspect-level or overall scores?
>
> **A5:** For *non-MJ-VIDEO methods*, we use the original model architectures and released weights, which do not involve aggregation of fine-grained scores. For *MJ-VIDEO methods*, **the aggregation process follows Eq.1 and Eq.2.** Specifically, we introduce additional Mixture-of-Experts (MoE) routing layers to weight the fine-grained scores during aggregation. Given the 28 fine-grained criteria scores $C \in \mathbb{R}^{28}$, the scores associated with each of the five predefined aspects $\{U_i\}_{i=1}^{5}$ are normalized as:
>
> $$
> C[U_i] = \text{softmax}(g'(h)[U_i]) \odot f(h)[U_i],
> $$
>
> where $U_i$ denotes the indices of the criteria corresponding to aspect $i$, and $h$ is the hidden feature. The overall preference score (OS) is then computed by aggregating the criteria scores using the aspect-level routing scores $\text{AR} \in \mathbb{R}^{5}$:
>
> $$
> \text{OS} = \sum_{i=1}^{5} \left[ \sum_{t \in U_i} C[t] \right] \text{AR}[i].
> $$
>
> This routing-based aggregation enables the model to adaptively emphasize the most relevant aspects and criteria, improving the alignment and generalizability of preference modeling in video generation.
>
> ---
>
> > What is the inter-annotator agreement for each evaluation aspect?
>
> **A6:** Thank you for your valuable suggestion! We evaluated inter-annotator agreement across the five dimensions in our benchmark: Alignment, Safety, Fineness, Coherence & Consistency, and Bias & Fairness, using five human annotators.
>
> We report both **Cohen’s kappa (κ)** to measure agreement, and **Spearman’s rho (ρ)** to assess rank correlation of ratings. These results are based on the statistics of 5 human annotators, which are shown below:
>
> | **Aspect**               | **Cohen’s κ** | **Spearman’s ρ** |
> | ------------------------ | ------------- | ---------------- |
> | Alignment                | 0.62          | 0.71             |
> | Safety                   | 0.66          | 0.68             |
> | Fineness (Object Detail) | 0.58          | 0.64             |
> | Coherence & Consistency  | 0.61          | 0.70             |
> | Bias & Fairness          | 0.54          | 0.60             |
>
> These results indicate moderate to substantial agreement of MJ-BENCH-VIDEO, supporting the reliability of our annotations.
>
> In addition, we evaluated how well two SOTA LLMs — Gemini and GPT-4o — agree with our final human consensus scores. Specifically, We prompt the LLM to judge the video preference pairs to see whether it has the same preference  judgment with annotated label. The below results suggest that our annotations are reliable as a benchmark. We will include these results into our next updated version.
>
> | **Model**      | **Average Agreement Rate** | **Average Spearman’s ρ** |
> | -------------- | -------------------------- | ------------------------ |
> | Gemini | 72.4%                      | 0.67                     |
> | GPT-4o         | 76.1%                      | 0.71                     |

---

> > ### Comment · Reviewer_WCuD · 2025-08-04
> >
> > Thanks for the informative response. As stated in Section 2.2.1, SafeSora is part of the proposed dataset. Why is it considered an OOD evaluation?

---

> > > ### Author Response · Authors · 2025-08-04
> > > **Clarification on OOD benchmarks**
> > >
> > > Thank you for your careful reading!
> > >
> > > 1. We apologize for the confusion. We acknowledge that SafeSora-test is not an OOD benchmark. We intended to demonstrate its performance as an in-distribution test set, as our MJ-Video model was trained exclusively on the corresponding training split.
> > >
> > > 2. However, please note that GenAI-Bench is indeed an OOD benchmark, and our method outperforms existing methods there. Additionally, during the rebuttal period, we evaluated 100 samples generated by Sora and ltx-video[1], which are not included in our training set and whose video distributions differ substantially from our training data. The results are as follows:
> > >
> > >
> > > | **Reward Model**      | **Sora (%)** | **ltx-video (%)** |
> > > |-------------------|----------|---------------|
> > > | Video-Score       | 73       | 66            |
> > > | Human Annotators  | 85       | 80            |
> > > | **MJ-VIDEO**          | **79**       | **72**            |
> > >
> > > These results show that MJ-VIDEO can still provide accurate and reliable judgments for OOD videos produced by these SOTA video generators, demonstrating its effectiveness as a general-purpose judge for video generation tasks.
> > >
> > > 3. We are now evaluating results on VisionReward[2] and Rapidata/text-2-video-human-preferences-veo3 at huggingface, two additional OOD benchmarks, and we will update these results very soon.
> > >
> > > We appreciate your patience and understanding.
> > >
> > > [1] HaCohen, Yoav, et al. "Ltx-video: Realtime video latent diffusion" arXiv preprint arXiv:2501.00103 (2024).
> > >
> > > [2] Xu, Jiazheng, et al. "Visionreward: Fine-grained multi-dimensional human preference learning for image and video generation" arXiv preprint arXiv:2412.21059 (2024).

---

> ### Author Response · Authors · 2025-08-05
> **OOD Results on VisionReward and VEO3**
>
> Thank you for your patience.
>
> We have now evaluated MJ-VIDEO and VideoScore on two additional out-of-distribution (OOD) benchmarks: **VisionReward** and **Rapidata/text-2-video-human-preferences-veo3**, both hosted on Hugging Face. To ensure fairness and efficiency, we randomly sampled a subset of examples from each dataset for evaluation. The average preference prediction accuracies are summarized below:
>
> | **Benchmark**                                                | **MJ-VIDEO Accuracy** | **VideoScore Accuracy** |
> | ------------------------------------------------------------ | --------------------- | ----------------------- |
> | VisionReward                             | **56.5**              | 54.9                    |
> | text-2-video-human-preferences-veo3  | **60.7**              | 58.8                    |
>
> These results further validate the robustness and generalization ability of MJ-VIDEO in aligning with human preferences on diverse and challenging OOD datasets.

---

> > ### Comment · Reviewer_WCuD · 2025-08-05
> >
> > Thanks for the new results. They look good. I will increase my score.

---

> > > ### Author Response · Authors · 2025-08-05
> > > **Thank you for your recognition!**
> > >
> > > We are happy to see our response were helpful to address your concerns. Thank you very much for your efforts reviewing our paper and the decision to increase the score!

---

### Official Review · Reviewer_NVDu · 2025-07-02

**Clarity:** 3
**Significance:** 4
**Originality:** 2
**Rating:** 5
**Confidence:** 4

**Summary:**

This paper presents a benchmark (MJ-Bench-Video) for fine-grained preferences in text-to-video models. Building upon this preference data, the paper also introduces MJ-Video, a MoE style video reward model that provides rewards for several aspects of the video/instruction. Finally, the paper also demonstrates that incorporating the generated preferences also provides notable improvements especially when compared to prior work on preference-tuning text-to-video models.

**Questions:**

I find the aspect of reusing image data as a source for this benchmark curious: The annotation protocol simply uses an image-to-video model to extend the images into videos. I am curious if the goal behind this was to specifically collect preference data for the image-to-video task as opposed to text-to-video (since they are slightly different in their requirements)? However, all the downstream use cases of the preference data seem to be directly for text-to-video tasks.

I'm also curious about the results in Tab. 3; while the improvements in the Human Eval are quite noticeable, the gains in the VBench AutoEval seem a lot more limited. Would the authors have an explanation for why this happens?

**Ethical Concerns:**

["NO or VERY MINOR ethics concerns only"]

**Final Justification:**

After looking at all the reviews and the rebuttal, it is fairly clear that the paper makes good contributions and is ready for acceptance.

**Limitations:**

Yes

**Quality:**

4

**Strengths And Weaknesses:**

Strengths:

The paper makes very strong contributions in my opinion. As far as I can understand, this is one of the most comprehensive sources of data for preference learning in video generation. Further, the paper also demonstrates that the proposed reward model (MJ-Video) is able to outperform even models like GPT-4o comprehensively. Finally, using the reward model to improve the underlying text-to-video models also reiterates the strength of the overall contributions.

Weaknesses:

The main weakness in the paper is that the choice of models it relies upon are somewhat outdated (e.g. the preference data is using Stable Video Diffusion for Image-to-Video, while the other models used for generating data are OpenSora, VideoCrafter etc. which are far away from both closed-source frontier models such as Sora, Veo, Kling etc. but also open-source models such as Wan, HunyuanVideo, Mochi). Incorporating superior models for generating preference data would certainly increase the utility of the MJ-Bench-Video in the long run.

---

> ### Author Rebuttal · Authors · 2025-07-30
>
> Thank you for your detailed review and helpful suggestion on our paper!  We would like like to respond to your conerns point-to-point.
>
>
> > The main weakness in the paper is that the choice of models it relies upon are somewhat outdated (e.g. the preference data is using Stable Video Diffusion for Image-to-Video, while the other models used for generating data are OpenSora, VideoCrafter etc. which are far away from both closed-source frontier models such as Sora, Veo, Kling etc. but also open-source models such as Wan, HunyuanVideo, Mochi). Incorporating superior models for generating preference data would certainly increase the utility of the MJ-Bench-Video in the long run.
>
> **A1:** Curating high-quality, human-annotated preference data is a time-consuming process that requires a stable generation pipeline. At the time of submission, we relied on well-established open-source models to ensure the smooth progress of the annotation process. Actually, we are continuously updating our benchmark, incorporating more high-quality videos—mainly generated by Sora—into the publicly released version. These updated results will be reported in the next updated version, and we encourage the community to use and contribute to the evolving online benchmark.
>
> In addition, wo showed that our MJ-VIDEO model can perform well on judging veideos (100 samples) generated by Sora or recent ltx-video[1] as follows:
>
> | **Reward Model**      | **Sora (%)** | **ltx-video (%)** |
> |-------------------|----------|---------------|
> | Video-Score       | 73       | 66            |
> | Human Annotators  | 85       | 80            |
> | **MJ-VIDEO**          | **79**       | **72**            |
>
> This indicates that most recent models could still benefit from our MJ-VIDEO and MJ-BENCH-VIDEO for alignment.
>
>
> ---
>
> > I find the aspect of reusing image data as a source for this benchmark curious: The annotation protocol simply uses an image-to-video model to extend the images into videos. I am curious if the goal behind this was to specifically collect preference data for the image-to-video task as opposed to text-to-video (since they are slightly different in their requirements)? However, all the downstream use cases of the preference data seem to be directly for text-to-video tasks.
>
> **A2:** First, the image-to-video process heavily relies on the text prompt, rather than solely on the image. This makes it a popular method for augmenting the limited amount of native video data used by WAN2.1[2], VIDEOSCORE[3], VideoCrafter2[4] and etc. The generated videos still closely align with the text prompt. Additionally, the image-to-video data only accounts for about 1/3 data, while the majority data (2/3) comes from either existing native video preference datasets or directly generated text-to-video data. Furthermore, through our human annotation process, we ensure that each video closely aligns with the input prompt. Low-quality videos are either labeled as dispreferred samples or discarded entirely.
>
> ---
>
> > I'm also curious about the results in Tab. 3; while the improvements in the Human Eval are quite noticeable, the gains in the VBench AutoEval seem a lot more limited. Would the authors have an explanation for why this happens?
>
> **A3:** First, VBench[5] metrics emphasize frame‑level attributes (e.g., motion, scene consistency) and may not fully capture nuanced alignment improvements as perceived by human judges. In contrast, human evaluation focuses on overall quality and text-to-video alignment, where MJ-VIDEO’s fine-grained feedback provides better support for assessing text-to-video generation.
>
> Another reason is that the human evaluation protocol follows the annotation standard of MJ-BENCH-VIDEO, which offers a more comprehensive assessment of text-to-video alignment than VBench. This may also benefit models trained on MJ-VIDEO, as the training data and evaluation share the same protocol. Note that MJ-VIDEO still outperforms baseline methods on out-of-domain benchmarks which use different evaluation protocol, as shown in Table 2.
>
>
> [1] HaCohen, Yoav, et al. "Ltx-video: Realtime video latent diffusion." arXiv preprint arXiv:2501.00103 (2024).
>
> [2] Wan, Team, et al. "Wan: Open and advanced large-scale video generative models." arXiv preprint arXiv:2503.20314 (2025).
>
> [3] He, Xuan, et al. "Videoscore: Building automatic metrics to simulate fine-grained human feedback for video generation." arXiv preprint arXiv:2406.15252 (2024).
>
> [4] Chen, Haoxin, et al. "Videocrafter2: Overcoming data limitations for high-quality video diffusion models." Proceedings of the IEEE/CVF Conference on Computer Vision and Pattern Recognition. 2024.
>
> [5] Huang, Ziqi, et al. "Vbench: Comprehensive benchmark suite for video generative models." Proceedings of the IEEE/CVF Conference on Computer Vision and Pattern Recognition. 2024.

---

> > ### Comment · Reviewer_NVDu · 2025-08-03
> >
> > I thank the authors for providing the clarifications, and I'm happy to maintain my accept rating.

---

> > > ### Author Response · Authors · 2025-08-04
> > > **Thank you for your recognition!**
> > >
> > > We are happy to see our clarifications were helpful. Thank you for your valuable suggestion and support!

---

### Official Review · Reviewer_uVYB · 2025-07-03

**Clarity:** 3
**Significance:** 3
**Originality:** 3
**Rating:** 5
**Confidence:** 4

**Summary:**

The work introduces MJ-Video-Bench and MJ-Video. MJ-Bench-Video is a benchmark for preference alignment of video models in 5 main categories 28 attributes. MJ-Video is a MoE based video reward model that can provide scores for the 5 main categories and the 28 attributes. The authors uses both existing and newly created data to create the benchmark. The data goes through a filteration and an annotation process to get the final benchmark. The MoE model goes through three stage of training. The results are shown for proposed benchmark and existing benchmark.

**Questions:**

I'm willing to change the score based on the rebuttal response. Mainly I have one question. I would appreciate it if you could provide more information regarding the cost per a video that ended up in the final benchmark.

**Ethical Concerns:**

["NO or VERY MINOR ethics concerns only"]

**Final Justification:**

I thank the reviewers for the response on cost. The cost stated by authors is much lower than I expected. I'm keeping the same score: 5: accept

**Limitations:**

yes

**Paper Formatting Concerns:**

No errors found

**Quality:**

3

**Strengths And Weaknesses:**

**Strengths**

* The work is important given the rise of video generation models and due to the lack of comprehensive preference alignment reward models.

* The MJ-Bench-Video is comprehensive and well-thought.

* The benchmark creation process is thorough and human contribution is appreciated due to the low performance of current video models.

* Choosing MoE as the reward model is good and the implementation seems well thought. The three stage training ensures the alignment between 28 criterias and 5 aspects.

* Experiments are shown for the proposed benchmark and existing benchmarks. Variety of models chosen are extensive.

* The limitations are well described by authors

**Weakness**

* Heavily relied on GPT-4 for video analysis etc which would have made the whole process very expensive. While human evaluations and annotations guaranteed more reliable outcome, the benchmark creation pipeline cannot be scalable nor cost effective.

* There is no information regarding the cost for creating the benchmark or average resource consumption for one video that ended up in the benchmark.

---

> ### Author Rebuttal · Authors · 2025-07-30
>
> Thank you for your recognition of our paper! We would like like to respond to your conerns point-to-point.
>
> > Heavily relied on GPT-4 for video analysis etc which would have made the whole process very expensive. While human evaluations and annotations guaranteed more reliable outcome, the benchmark creation pipeline cannot be scalable nor cost effective.
>
> **A1:** We acknowledge that the GPT-involved benchmark curation process can become costly. However, using an LLM-involved pipeline may currently be the most effective method for scaling. To ensure the accuracy of human preference, we still have to rely on human annotators for preference labeling - a task that currently cannot be replaced by a LLM (as shown by the judging performance of GPT-4o or even SOTA large reasoning models such as OpenAI's O1 in Table 1 and 11, respectively).
>
> ---
>
> > There is no information regarding the cost for creating the benchmark or average resource consumption for one video that ended up in the benchmark.
> > I'm willing to change the score based on the rebuttal response. Mainly I have one question. I would appreciate it if you could provide more information regarding the cost per a video that ended up in the final benchmark.
>
> **A2:** Thank you for your constructive suggestion. We will include the below table into our next updated version. Here we stat the breakdown costs for setting up our benchmark (cost per sample) as follows:
>
> | Component | Description | Estimated Cost (USD) |
> | --------------------------------------- | ----------------------------------------------------------| -------------------- |
> | Human annotation (5 annotators × 2 min) | 2 minutes total; $1/hr | 0.16 |
> | Automated quality check (LLM+heuristic) | Lightweight review using Gemini Flash-Lite | 0.03 |
> | Spot sampling for manual audit | Periodic sampling for quality assurance | 0.01 |
> | Total | Total cost per video sample | 0.2 |
>
>
> We can see that the cost is relatively low and acceptable for such fine-grained annotation.

---

> ### Comment · Area_Chair_L8py · 2025-08-06
> **Comment by AC**
>
> Dear Reviewer uVYB,
>
> Thank you for your participation in the review process. Please engage in the discussion phase by following these guidelines:
>
> - Read the author rebuttal;
> - Engage in discussions;
> - Fill out the "Final Justification" text box and update the "Rating" accordingly.
>
> The deadline is Aug 8, 11.59pm AoE.
>
> Thanks,
>
> AC

---

### Decision · Program_Chairs · 2025-09-17

**Decision:**

Accept (spotlight)

**Comment:**

The paper introduces MJ-Video-Bench, a benchmark dataset for video preference alignment, and MJ-Video, a video reward model. After the rebuttal, all reviewers voted for acceptance. The AC agrees with the reviewers that this paper makes a strong contribution, especially in the domain of video generation, as good open-source datasets and reward models for post-training optimization of video generation models are lacking.